# Predicting a Favorable (mRS 0–2) or Unfavorable (mRS 3–6) Stroke Outcome by Arterial Spin Labeling and Amide Proton Transfer Imaging in Post-Thrombolysis Stroke Patients

**DOI:** 10.3390/jpm13020248

**Published:** 2023-01-29

**Authors:** Qinmeng He, Guomin Li, Meien Jiang, Qianling Zhou, Yunyu Gao, Jianhao Yan

**Affiliations:** 1Guangdong Second Provincial General Hospital, School of Medicine, Jinan University, Guangzhou 520632, China; 2Department of Radiology, Guangdong Second Provincial General Hospital, Guangzhou 510317, China; 3Central Research Institute, United Imaging Healthcare, Shanghai 201807, China

**Keywords:** magnetic resonance imaging, amide proton transfer, arterial spin labeling, subacute phase ischemic stroke

## Abstract

(1) Background: The objective of this study was to determine whether arterial spin labeling (ASL), amide proton transfer (APT), or their combination could distinguish between patients with a low and high modified Rankin Scale (mRS) and forecast the effectiveness of the therapy; (2) Methods: Fifty-eight patients with subacute phase ischemic stroke were included in this study. Based on cerebral blood flow (CBF) and asymmetry magnetic transfer ratio (MTRasym) images, histogram analysis was performed on the ischemic area to acquire imaging biomarkers, and the contralateral area was used as a control. Imaging biomarkers were compared between the low (mRS 0–2) and high (mRS 3–6) mRS score groups using the Mann–Whitney U test. Receiver operating characteristic (ROC) curve analysis was used to evaluate the performance of the potential biomarkers in differentiating between the two groups; (3) Results: The rAPT 50th had an area under the ROC curve (AUC) of 0.728, with a sensitivity of 91.67% and a specificity of 61.76% for differentiating between patients with low and high mRS scores. Moreover, the AUC, sensitivity, and specificity of the rASL max were 0.926, 100%, and 82.4%, respectively. Combining the parameters with logistic regression could further improve the performance in predicting prognosis, leading to an AUC of 0.968, a sensitivity of 100%, and a specificity of 91.2%; (4) Conclusions: The combination of APT and ASL may be a potential imaging biomarker to reflect the effectiveness of thrombolytic therapy for stroke patients, assisting in guiding treatment approaches and identifying high-risk patients such as those with severe disability, paralysis, and cognitive impairment.

## 1. Introduction

Stroke is one of the most frequent causes of disability or death in the world [1]. Ischemic stroke, which is typically caused by blood vessel occlusion, is the most common type of stroke [2]. When it occurs, there is a lack of cerebral blood flow (CBF) [3] and a decrease in the pH of brain tissues [4]. Prior studies have demonstrated that tissue pH [5] and CBF [6] values can reflect functional and structural damage to ischemic brain tissue. Clinically, there are many patients admitted to the hospital for subacute phase ischemic stroke beyond the “golden hour (≤4.5 h)” [7], and although patients in this phase have exceeded the optimal thrombolysis time, studies have shown that there is still an ischemic penumbra in subacute phase ischemic stroke [8,9,10]. Therefore, it is of good clinical significance to study the blood perfusion and pH of subacute phase ischemic stroke for understanding the pathophysiological mechanism and guiding clinical treatment. Thrombolytic therapy is widely used in the treatment of patients with ischemic stroke [11], but symptoms may worsen even when the therapy is successful [12]. Clinically, neurologists often use the Modified Rankine scale (mRS), a scale with a superficial evaluation process, to assess the prognosis of patients with cerebral infarction. Comorbidities and the socioeconomic level of the patient may have an impact on the mRS score indirectly [13]. In this way, seeking true anticipation of the effects after thrombolytic therapy is an important part of clinical work.

Cranial MRI is essential to guide reperfusion therapy in ischemic stroke. T1-weighted liquid attenuation inversion recovery (T1-FLAIR) imaging, T2-FLAIR imaging, and diffusion-weighted imaging (DWI) are commonly used clinically to obtain information on the volume of ischemic lesions [14]. DWI is sensitive to early cytotoxic edema in ischemic stroke because it recognizes changes in the self-diffusion of water molecules, so it is commonly used clinically to determine the presence of cerebral infarction. DWI combined with T1-FLAIR imaging and T2-FLAIR imaging can stage cerebral infarction. However, with the discovery of the study, the abnormal signal range shown by DWI as the core site of infarction is not accurate, and studies and clinical prognosis follow-ups have found that DWI overestimates the core range of infarction, which leads to differences in the scope of the final judgment of salvable tissue [15]. In addition, although national stroke guidelines recommend DWI for stroke diagnosis, associated ischemic lesions have not been detected in one-third of patients with non-disabling stroke [16]. Therefore, it is important to find advanced imaging methods that accurately reflect the early changes in ischemic stroke.

Recently, advanced MRI techniques, such as arterial spin labeling (ASL) [16] and amide proton transfer (APT) imaging [17,18], have been applied to clinical research on strokes. ASL is a technique that provides perfusion imaging without the use of exogenous contrast agents. By magnetically labeling arterial blood as an endogenous contrast, ASL allows the quantitative measurement of CBF [19]. DWI is widely used as a well-established perfusion-weighted imaging (PWI) technique to evaluate clinical applications in early cerebral ischemia (ACI) [14]. Studies have shown that ASL can detect abnormal perfusion of cerebral blood flow earlier than DWI [20] and can have good consistency with DWI-PWI technology (DWI-PWI), as well accurately assess the degree, extent, and lesion of perfusion reduction. In the diagnosis of ACI, 3D-ASL has a better role in showing infarct lesion extent and perfusion characteristics [20,21]. Moreover, ASL technology can effectively detect the ischemic semi-dark band in patients with ACI, and can be generalized to patients with clinical suspicion of ACI [21]. For patients with ACI after thrombolytic therapy, the 3D-ASL sequence can also timely reflect the recovery of blood flow reperfusion in the ischemic infarction area, find out whether there is an abnormal perfusion area, which is conducive to the comprehensive assessment of brain tissue blood flow perfusion information [21], and predict the occurrence of ischemia-reperfusion injury [22].

APT imaging enables the detection of amide protons of mobile intracellular proteins and peptides by exploiting the chemical exchange saturation transfer (CEST) effect. Because the exchange rate is base-catalyzed, APT imaging is sensitive to pH values [23]. MRI structural imaging is difficult to show the acid-base metabolism changes of cerebral infarction, and the current magnetic resonance imaging (MRS) technology can detect the changes of brain metabolites and measure the pH value of tissues, but its spatial resolution is low and its application is more limited [24]. The use of amide proton transfer imaging (APT), as a non-invasive imaging method, for the assessment of tissue metabolism changes after cerebral infarction, the evolution of ischemic tissue chemical environment can be simulated, the pH of the in vivo environment, the concentration of proteins and peptides can be inferred, the changes of brain metabolites can be quickly diagnosed, the metabolism of cerebral infarction brain tissue can be evaluated in time, and the evolution of the ischemic tissue chemical environment can be analyzed in time. APT technology has developed rapidly in ischemic stroke and some progress has been made in both animal experiments and clinical studies. Studies have shown that when reperfusion is re-perfusion after occlusion, the APT value will gradually increase and the acidosis area will gradually decrease [25], while the lactate concentration is inversely correlated with the APT value, which indicates that APT is a PH-weighted imaging technique that shows tissue metabolism and ischemia reversal in perfusion therapy, and can be qualitatively analyzed to evaluate the prognostic effect.

Although ASL and APT may provide useful information about the stroke microenvironment, due to the complex pathophysiological and metabolic changes following cerebral infarction, there is still no uniform evaluation standard, and it remains to be seen whether their use alone or in combination can accurately predict the prognosis after thrombolytic therapy. This study evaluated the clinical value of APT, ASL, and their combination in differentiating between patients with low and high mRS scores and in predicting the prognosis of forecasting the effectiveness of the therapy or the patient’s response to the therapy in subacute phase ischemic stroke. A comparison was made between the performance of ASL, APT, and their combination to identify the optimal biomarker.

## 2. Materials and Methods

### 2.1. Subject

This study was approved by the ethics committee of our hospital. Written informed consent was obtained from all participants. Patients with a preliminary clinical diagnosis of subacute phase ischemic stroke between May 2020 and May 2022 were included in the study. Other inclusion criteria were as follows: (1) stroke in patients meeting the criteria set by the Chinese Guidelines for the Diagnosis and Treatment of Acute Ischemic Stroke 2020 [26]; (2) patients with no previous strokes or those with good recovery after the strokes; (3) age ≥18 years; (4) within 2–10 days from the onset of stroke; (5) the infarct has a definite hyperintensity in DWI; (6) availability of mRS before and after thrombolytic therapy; and (7) undergoing intravenous thrombolytic therapy. The exclusion criteria were as follows: (1) patients who were unable to undergo MRI examination; (2) lesions with a stroke less than 10 mm in diameter on DWI images; (3) patients with a history of cranial surgery; (4) patients with cerebral hemorrhage, subarachnoid hemorrhage, brain tumors and craniocerebral trauma, postoperative, psychiatric diseases; and (5) poor image quality due to severe motion artifacts. Clinical data, surgical history, and laboratory findings were recorded for all the patients. Stroke severity was assessed using the National Institutes of Health Stroke Scale (NIHSS) [23] on admission and functional outcomes were assessed 30 days after stroke onset using the modified Rankine Scale (mRS) [27]. Patients with favorable outcomes after thrombolysis were mRS 0–2 and those with unfavorable outcomes were mRS 3–6 [27]. The NIHSS and mRS numbers were assessed by two experienced neurologists based on the patient’s medical records. Any disagreement was resolved by consensus.

### 2.2. MRI Study

All patients were scanned after the first day of thrombolysis [28] on a 3T MRI scanner (uMR 780, United Imaging Healthcare, Shanghai, China) with a 32-channel head coil. Imaging scans ranged from the parietal skull to the foramen magnum. All patients received a routine MRI examination, including T1-FLAIR, T2-FLAIR, and DWI. APT and ASL MRI examinations were also performed. A routine MRI examination is done to determine the site of cerebral infarction and rule out other diseases. T1-weighted FLAIR (The voxel size = 0.69 × 0.63 × 5.00 mm^3^, the field of view = 230 mm, repetition time = 2221 ms, echo time = 10.2 ms, inversion time = 960 ms, 2 min 27 s); T2-weighted FLAIR (The voxel size = 1.00 × 0.80 × 5.00 mm^3^, the field of view = 230 mm, repetition time = 8000 ms, echo time = 108.48 ms, inversion time = 2500 ms), DWI (The voxel size = 1.31 × 1.31 × 5.00 mm^3^, the field of view = 230 mm, four averages, b = 0 and 1000 s/mm^2^, repetition time = 2894 ms, echo time = 97.1 ms, 21 slices, 1 min 47 s) with apparent diffusion coefficient (ADC) calculation. APT imaging was performed using a 2D single-shot fast spin echo (SSFSE) sequence in single-slice with the following parameters: The voxel size = 1.80 × 1.80 × 8.00 mm^3^, the field of view = 230 mm, repetition time = 4500 ms, echo time = 59.62 ms, 2 min 25 s). The slice position was selected to include the maximum level of the lesion. The APT signal was represented by the magnetic transfer ratio asymmetry (${\mathrm{MTR}}_{\mathrm{asym}}$), which was calculated as:$${\mathrm{MTR}}_{\mathrm{asym}}\left(3.5\mathrm{ppm}\right)=\left(s\left(-3.5\mathrm{ppm}\right)-s\left(+3.5\mathrm{ppm}\right)\right)/s_{0}$$
where (±3.5 ppm) is the signal with radio frequency (RF) pulse shift ±3.5 ppm, and $s_{0}$ represents the signal without application of off-resonance RF [29]. ASL images were acquired using a 3D pseudo-continuous arterial spin labeling sequence with a 3D GRASE readout. Sequence parameters are listed as follows: voxel size = 3.50 × 3.50 × 6.00 mm^3^, the field of view = 224 × 224 mm^2^, repetition time = 5500 ms, echo time = 13.58 ms, labeling duration = 1800 ms, post-labeling delay = 2000 ms, and the total acquisition time = 3 min and 40 s. CBF was determined by [30]:$$\mathrm{CBF}=\frac{\Delta \mathrm{Mexp}\left(w/T_{1a}\right)\exp\left(\mathrm{TE}/T_{2a}\right)}{{\rho M}_{0a}2{\alpha T}_{1a}}$$
where ΔM is the difference between the control and labeled image intensities, w is the post-labeling delay, $T_{1a}$ is the T1 of arterial blood, TE is the echo time, $T_{2a}$ is the T2 of arterial blood, ρ is the density of the brain tissue, α is the labeling efficiency, and $M_{0a}$ is the equilibrium magnetization of the arterial blood, which can be calculated by:$$\mathrm{lambdas}=\frac{S_{\mathrm{csf}}\lambda_{a}}{1-\exp\left(-\frac{\mathrm{TR}}{T_{1\mathrm{csf}}}\right)}$$
where $S_{\mathrm{csf}}$ is the signal intensity of cerebrospinal fluid in a manually defined ventricular region, $\lambda_{a}$ is the volume of water per ml of arterial blood (0.76), TR is the sequence repetition time, $T_{1\mathrm{csf}}$ is the relaxation rate of CSF.

### 2.3. Images Analysis

The ischemic areas were identified using DWI. We selected the largest layer of cerebral infarction lesions on DWI and obtained APT images of the same layer. As illustrated in Figure 1, regions of interest (ROIs) were manually chosen on DWI by two experienced neuroradiologists without information on the patients’ pathological diagnosis and mRS score, then the lower resolution images (APT and CBF) were registered to the higher resolution image (DWI) automatically using image registration software (FSL). Any disagreement was resolved by consensus. ROI 1 was drawn on the ischemic areas, and ROI 2 was placed on the normal contralateral hemisphere. The ROIs did not include areas of necrosis, hemorrhage, or cystic degeneration. Histogram analysis was performed on ischemic and the contralateral areas to obtain imaging parameters, including the mean, max, min, kurtosis, skewness, and percentile value for the 10th, 25th, 50th, 75th, and 90th percentiles. The difference between the ischemic and the normal contralateral areas (ROI 1-ROI 2) was calculated to obtain rASL and rAPT. In this study, patients with an mRS (0–2) were classified into the good prognosis group, and the rest were classified into the poor prognosis group. All imaging parameters for rASL and rAPT were compared between the two groups using non-parametric Mann–Whitney U tests. Parameters with significant differences between the two groups were selected as imaging markers. The least absolute shrinkage and selection operator (LASSO) was then applied to select imaging markers using a 10-fold cross-validation approach. Finally, the selected markers were combined using logic regression. The selected markers and their combinations were analyzed using receiver operating characteristic (ROC) curves and the area under the curve (AUC) was calculated to evaluate the performance of the selected parameters. Statistical significance was set at *p* < 0.05. All the analyses were performed using MedCalc version 20 (MedCalc Software, Mariakerke, Belgium) and MATLAB (MathWorks, Natick, MA, USA).

## 3. Results

### 3.1. Clinical Characteristics

A total of 73 patients were enrolled early, of whom five were affected by motion artifacts, four due to claustrophobia, and six were excluded due to complications affecting neurological scores and finally, 58 cases (mean age 60.48 ± 11.12 years), including 43 males (69.63 ± 11.72 years) and 15 females (60.07 ± 9.56 years), met the inclusion criteria and were included in the subsequent analysis. Among all ischemic stroke patients, 53 had hypertension (91.38%), seven had diabetes mellitus (12.07%), 30 had hyperlipidemia (51.72%), and 15 had gouty arthropathy (25.86%). The demographic characteristics of the patients in the two groups are summarized in Table 1. There were no significant differences in age, sex, hypertension, diabetes mellitus, hyperlipidemia, and gouty arthropathy between the two groups. The NHISS and mRS (before the stroke onset) have significant differences. Table 2 shows the sample size needed in this study.

The mean ± SD of the stroke dimension is 3.09 ± 4.88. The mean ± SD of the mRS before the stroke onset is 3.22 ± 1.23. The mean time to symptom onset was 115 h and 50 min (about 4.8 days), ranging from 49 h 30 min to 238 h and 47 min (about 2 to 10 days).

### 3.2. Image of Stroke Patients

The DWI, ASL, and APT images of four patients are shown in Figure 2. The mRS scores of patients A, B, C, and D were 0, 3, 1, and 3, respectively, and the corresponding NIHSS scores upon administration were 5, 16, 7, and18, respectively. It can be seen from the ASL images that cerebral perfusion in the ischemic area was higher than that in the normal contralateral hemisphere in patient A and patient C, and perfusion in patients B and D ischemic areas is lower than in the normal contralateral hemisphere.

### 3.3. Statistical Analysis

All imaging parameters of rASL and rAPT acquired from the histogram analysis are presented in Table 2. It can be noted that the absolute values of most of the rAPT parameters in groups with mRS (0–2) were smaller than those in groups with mRS (3–6). In addition, the rASL parameters in the group with mRS (3–6) were significantly higher than those in the other groups. Table 3 shows the non-parametric Mann–Whitney U test results of all histogram parameters between the group with mRS (0–2) and the group with mRS (3–6). Four histogram parameters of rAPT were significantly different between the two groups, including the mean (*p* = 0.005), maximum (*p* = 0.011), and percentile values for the 50th (*p* = 0.019) and 90th (*p* = 0.045) percentiles. The results of multivariate statistical comparison are shown in Table 4. The ASL and APT parameters are significantly different from the NIHSS on admission, the mRS on admission, and the dimension of the stroke lesion

All parameters of rASL in the group with an mRS (0–2) were significantly higher than those in the other groups, except kurtosis and skewness. After a 10-fold cross-validation approach, LASSO, the mean and max of rAPT, and the min and max 90th percentiles of rASL were selected and analyzed using the ROC curve. The results of the ROC analysis are illustrated in Figure 3 and summarized in Table 5. rAPT 50th had an AUC of 0.728 with a sensitivity of 91.67% and a specificity of 61.76%. Moreover, the AUC, sensitivity, and specificity of the rASL max were 0.926, 100%, and 82.4%, respectively. Combining the parameters with logistic regression could further improve the performance in predicting prognosis, leading to an AUC of 0.968, a sensitivity of 100%, and a specificity of 91.2%.

## 4. Discussion

This study investigated the relationship between the mRS scores of patients after thrombolytic therapy and imaging biomarkers, including APT, ASL, and their combination. The results indicate the feasibility of APT, ASL, and their combination in predicting the therapy efficacy of subacute phase ischemic stroke patients after thrombolytic therapy. Both rASL and rAPT in the good prognosis group were significantly different from those in the poor prognosis group, indicating their clinical value in predicting the effectiveness of the therapy. The combination of rASL and rAPT had the highest AUC in predicting outcomes and might be a potential imaging biomarker for predicting stroke prognosis after thrombolytic therapy.

The rAPT was significantly different between the two groups, indicating that it could be used to predict the therapeutic efficacy of subacute phase ischemic stroke. This is consistent with previous studies which have shown that APT MRI can detect tissue pH [31,32]. When the cerebral artery is occluded, the corresponding blood perfusion pressure will gradually decrease, a series of pathophysiological changes will occur in the human body, the lactic acid concentration in the lesion area will gradually increase, the pH value will decrease, and the proton exchange rate of saturated amide and water will also be significantly reduced. In addition, studies using animal models have found that the study has found that the emergence of APT effect, the reduction of pH value will cause the slowdown of the exchange rate of amide protons and hydrogen protons in water, while the MTR in the brain tissue of dead rats will gradually decrease, and the calibration of APT in cerebral ischemia models has found that there will be obvious differences between PH values and normal brain tissues, which is basically consistent with the results of tissue staining behavior [18]. This may make rAPT an effective marker for understanding the outcome of subacute phase ischemic stroke. The mean rAPT value in the good prognosis group was approximately zero, and the standard error was low. These findings are also similar to those of previous studies [33]. Sui, H. J found that the ADC value of the cerebral infarction lesion area will gradually change from low to high with time [34], and in the subacute phase (days 2–10), as the acid-base balance of the perfusion area continues to alkalinize, the APTw reduction area shrinks, and the perfusion area improves correspondingly with the opening of the lateral branch circulation and reconstruction, and “false normalization” can occur. The average course of infarction cases in this group was 115 h and 50 min (about 4.8 days), from 49 h and 30 min to 238 h and 47 min (about 2–10 days), in the stage of acidosis to alkalinization change, resulting in microenvironments that affect APT values, such as PH concentration, which do not appear to change significantly. As a result, APT imaging cannot detect a significant difference between the stroke area and the contralateral normal area after thrombolysis. The mean value in the good prognosis group was lower than that in the poor prognosis group, which is inconsistent with previous research [35]. Studies have found that the APT signal of patients with different periods of ischemic stroke will be reduced to varying degrees, and the signal intensity in the hyperacute phase will be the lowest, which is basically consistent with the most severe cases of acidosis in the hyperacute ischemic area. With the prolongation of the onset time, acidosis will gradually decrease, and the signal intensity of APT can reflect the characteristics of PH value of ischemic tissues at different stages and different time points, which has a more advanced predictive effect. PH imaging may be a crucial substitute imaging biomarker for acute ischemic stroke, as it reflects the metabolic health of the tissue. The initial pH adjustment results in a slightly higher APT imaging contrast change because the amide proton exchange rate is predominately base-catalyzed [36]. Besides, several things will interfere with APT in vivo imaging: The direct saturation impact of the water proton, also known as the spillover effect, will occur when the saturation pulse excites the amide proton because its magnetic resonance frequency is very close to the water center frequency. The field strength and equipment settings have a significant impact on the direct saturation effect, and low field strength will make this effect more pronounced. Between two protons that are spatially very close to one another, there occurs a relaxation effect. The nuclear Euclidean effect occurs when one proton is energized and saturated and transfers energy to the other proton, increasing their resonance signal (Nuclearoverhauserenhancement, NOE) [37,38]. These will deliver upsetting polarization move outcomes, which will influence the precision of Well-suited imaging results. Because the overflow impact changes evenly with the water community recurrence, MTRaysm can eliminate the impact of this impact on the Able sign. Nonetheless, the charge move impact of strong macromolecules is not evenly circulated at the middle recurrence of water. Likewise, the reverberation recurrence of a few aliphatic protons is right at −3.5 ppm, which will likewise create the NOE outcome, which will disrupt the exactness of MTRaysm boundaries [39]. There was also a statistically significant difference in rASL between the groups with good prognoses and those with poor prognoses. This is similar to previously described results [40]. In normal brain tissues, intracellular pH is regulated by active and passive mechanisms and kept at 7.2 [41]. When ischemia occurs, the blood supply to the brain decreases. Such a change in the microenvironment can be detected in the form of CBF using ASL. In the case of poor cerebral perfusion, the extrusion of CO2 from the cell is limited, and the glucose and oxygen supplies are reduced [42], causing a decrease in bicarbonate buffering capacity and the depletion of glycogen and phosphocreatine. In general, intracellular pH relies on cerebral perfusion, intracellular energy reserves time. Affected by ischemia, brain tissue undergoes anaerobic metabolism, leading to lactic acid accumulation and decreased pH [5,36]. Cerebral hypoperfusion and reduced bicarbonate buffering capacity worsen tissue acidification, which frequently results in cell death and tissue damage. In this study, patients with lower mRS scores after thrombolysis generally presented with hyperperfusion, and the mean rASL value in the poor prognosis group was significantly lower than that in the good prognosis group. When an ischemic stroke occurs, the blood supply to a part of the brain decreases, leading to brain damage. Such damage can be alleviated if thrombolytic therapy can establish reperfusion on time. Hyperperfusion is believed to occur after effective vascular recanalization and tissue reperfusion because local autoregulation may require a few days to compensate for the extra blood volume that tissues receive as a result of their initial maximal vasodilatation due to acute ischemia [43]. In contrast, patients with higher mRS scores after thrombolysis typically showed hypoperfusion in the ASL. A low rASL value after thrombolytic therapy may indicate ineffective reperfusion. This may be due to the failure of the thrombolytic or no-reflow effect. According to previous research [44,45], multiple factors, including microvascular obstruction, edema, and occlusion via the endothelium, could cause the no-reflow effect, even though the primary occlusion was resolved. Owing to this phenomenon, the tissue was unable to receive the nutritional support needed to recover sustainably, which might lead to more severe brain damage and a higher mRS score.

As mentioned above, the CBF value detected by ASL reflects the perfusion microenvironment, and the ${\mathrm{MTR}}_{\mathrm{asym}}$ acquired by APT imaging indicates the metabolic health of the tissue. Combining ASL and APT may provide comprehensive information and further understanding of the stroke lesion microenvironment and pathophysiology. In this study, the combination of ASL and APT led to further improvements in the AUC. Prior studies used DWI, ASL, and APT mismatching to identify the ischemic core and penumbra [32]. The perfusion deficit area might recover spontaneously, and ischemic cores are potentially reversible; however, further information about the metabolic milieu is needed to make a diagnosis [46]. Clinically, head MR is usually performed after thrombolysis to understand post-thrombolytic recanalization. However, in this study, the combination of ASL and APT led to further improvements in AUC, and the combination of APT and ASL may be a potential imaging biomarker that reflects the outcome of thrombolytic therapy in patients with subacute ischemic stroke, thereby helping to guide treatment and identify high-risk patients. It is possible to apply the MRI techniques in this study for pre-thrombolytic selection of the stroke patients who will gain clinical benefit from thrombolytic therapy.

The general data analysis of patients in the infarction group showed that there was no clear correlation between the clinical and laboratory test results and the APTw value, which may be because the measurement of clinical indicators and laboratory tests of patients were one-time measurement results, and did not coincide with the MRI examination time, and there was a certain error, which required further large-sample epidemiological research.

This study has several limitations. First, the patients did not have a baseline MRI scan before thrombolytic therapy because such reperfusion therapy is time-critical, whereas an MRI scan is time-consuming. Patients were not divided into anterior circulation and posterior circulation strokes for study and it will be our next research plan. Moreover, when patients with cerebral infarction were admitted for examination, some patients were in serious condition, and patients or their parents were more worried and nervous and could not accurately describe the last known well time. Because most cerebral infarction patients are elderly, they are less sensitive to changes in their bodies, so it is difficult to detect their own accurate onset time. Secondly, the sample size of the study was relatively small. Further studies with larger sample sizes should be conducted. Finally, APT imaging was acquired with only one slice using a 2D SSFSE sequence, whereas ASL imaging was performed using a 3D sequence.

## 5. Conclusions

In this study, the combination of APT and ASL may be a potential imaging biomarker to reflect the outcome of thrombolytic therapy for subacute phase ischemic stroke patients and thereby help to guide treatment approaches and identify high-risk patients, such as those with severe disability, paralysis, and cognitive impairment.

## Figures and Tables

**Figure 1 jpm-13-00248-f001:**
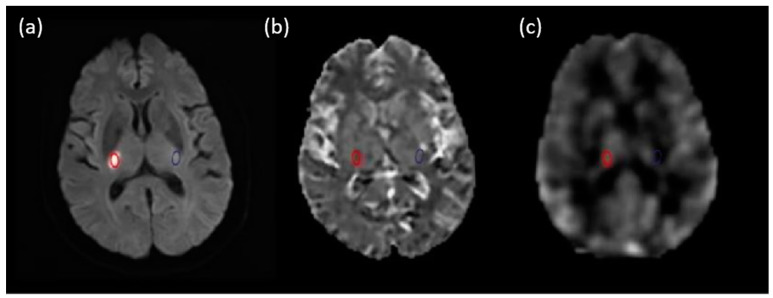
Illustration of ROI, which was placed on the lesion and marked with a red circle. (**a**) the reference DWI image (b = 1000). (**b**) MTRasym map. (**c**) CBF map. The contralateral to the lesion with a blue circle.

**Figure 2 jpm-13-00248-f002:**
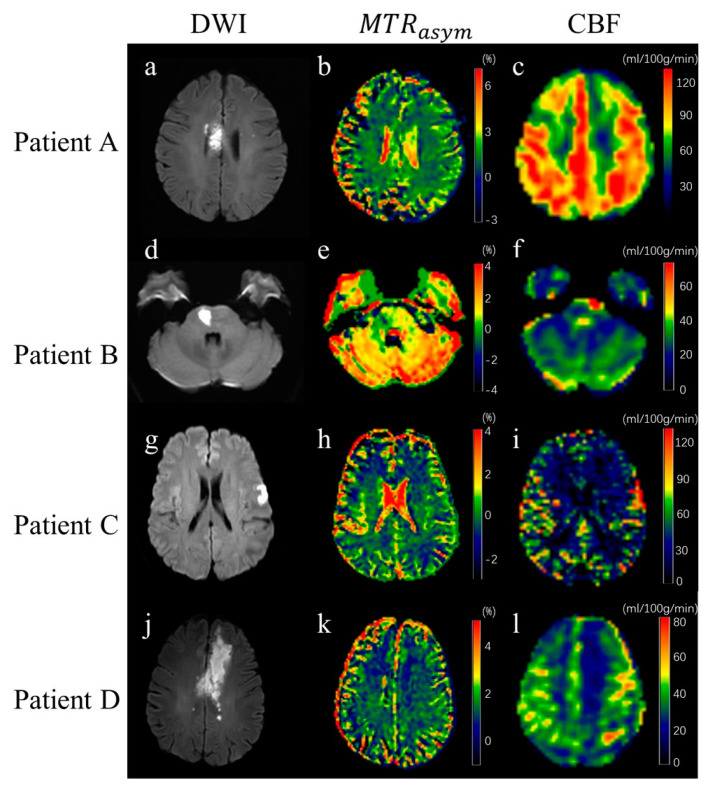
Rows 1–4 show an illustration of MR images of four patients. Columns 1–3 contain DWI images, ${\mathrm{MTR}}_{\mathrm{asym}}$ maps, and CBF maps, respectively.

**Figure 3 jpm-13-00248-f003:**
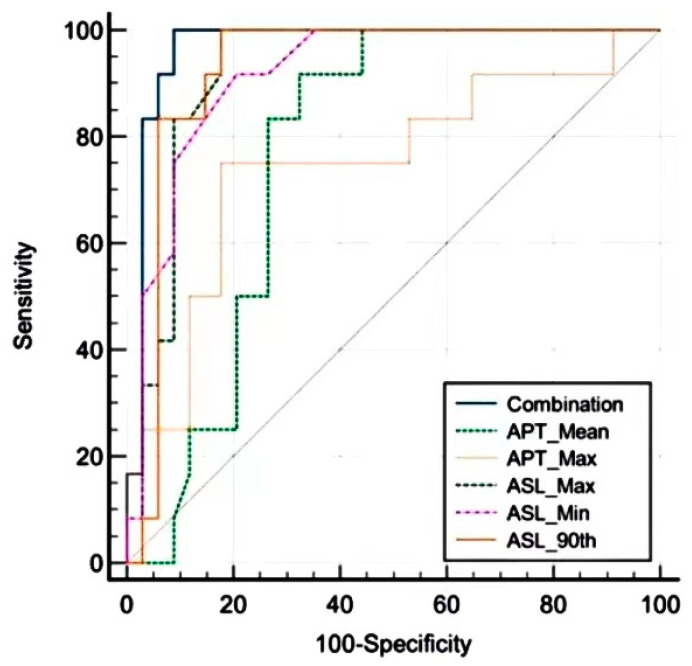
ROC analysis of APT, ASL, and their combination for predicting prognosis.

**Table 1 jpm-13-00248-t001:** The demographic and clinical characteristics of the group with mRS (0–2) and the group with mRS (3–6). Values are mean ± standard deviations.

Clinical Information	mRS (0–2)	mRS (3–6)	*p* Value
Sex (male: female)	13:5	30:10	0.823
Age (years)	62.2 ± 11.4	59.7 ± 11.1	0.196
NIHSS	10.8 ± 2.1	15.1 ± 3.1	<0.001 *
Hypertension	16(88.89%)	37(92.5%)	0.65
Diabetes mellitus	4(22.22%)	4(10.0%)	0.21
Hyperlipemia	11(61.11%)	19(47.5%)	0.34
Gouty arthropathy	4(22.22%)	11(27.5%)	0.67
mRS (admission)	1.8 ± 0.4	3.9 ± 0.8	<0.001 *

The marked with “*” is statistically significant (*p* < 0.05).

**Table 2 jpm-13-00248-t002:** The power values for sample size 58.

Parameter	Power
rAPT mean	0.6108
rAPT max	0.6465
rASL max	0.7779
rASL Min	0.7992
rASL 90th	0.7793

**Table 3 jpm-13-00248-t003:** The *p*-values for the comparison of histogram parameters between the group with low mRS and the other group. Values are mean ± standard deviations.

	rAPT(%)			rASL(mL/100 g/min)		
	mRS < 2	mRS ≥ 2	*p* Value	mRS < 2	mRS ≥ 2	*p* Value
Mean	0.14 ± 0.68	0.21 ± 0.73	0.005 *	27.45 ± 36.55	−7.86 ± 39.8	<0.001 *
Max	−0.39 ± 1.7	−1.34 ± 7.91	0.011 *	30.61 ± 36.44	−11.55 ± 79.45	<0.001 *
Min	0.46 ± 1.26	0.89 ± 2.37	0.881	18.94 ± 30.58	−6.6 ± 12.58	<0.001 *
Kurtosis	−0.59 ± 1.62	0.23 ± 2.84	0.076	−0.11 ± 0.49	−0.11 ± 0.78	0.635
Skewness	−0.17 ± 0.95	−0.03 ± 1.12	0.565	−0.25 ± 0.62	0.06 ± 0.77	0.532
10th	0.26 ± 0.86	0.48 ± 1.05	0.453	21.72 ± 31.83	−9.48 ± 14.95	<0.001 *
25th	0.17 ± 0.63	0.38 ± 0.73	0.094	25.56 ± 37.41	−10.73 ± 17.93	<0.001 *
50th	0.18 ± 0.72	0.32 ± 0.76	0.019 *	28.25 ± 39.63	−7.91 ± 37.39	<0.001 *
75th	0.08 ± 0.93	0.11 ± 0.8	0.054	29.63 ± 38.3	−5.57 ± 65.93	<0.001 *
90th	0.07 ± 1.17	−0.22 ± 2.21	0.045 *	31.04 ± 35.69	−8.42 ± 73.87	<0.001 *

The marked with “*” is statistically significant (*p* < 0.05).

**Table 4 jpm-13-00248-t004:** The results of multivariate statistical comparison.

Parameter	NIHSS on Admission	mRS on Admission	The Dimension
rAPT mean	0. 0096 *	0. 0024 *	0. 0071 *
rAPT max	0. 0068 *	0. 0055 *	0. 0038 *
rASL max	0. 0001 *	0. 0038 *	0. 0061 *
rASL Min	0. 0019 *	0. 0026 *	0. 0041 *
rASL 90th	0. 0004 *	0. 0084 *	0. 0071 *

The marked with “*” is statistically significant (*p* < 0.05).

**Table 5 jpm-13-00248-t005:** ROC curve analyses of the selected parameters.

Variable.	AUC	Sensitivity (%)	Specificity (%)	SE	95% CI
Combination	0.968	100	91.2	0.0258	0.869 to 0.998
rAPT mean	0.771	91.7	67.6	0.0679	0.623 to 0.882
rAPT max	0.75	75	82.4	0.0919	0.600 to 0.866
rASL max	0.922	100	82.4	0.0406	0.803 to 0.980
rASL min	0.918	100	88.2	0.0408	0.798 to 0.978
rASL 90th	0.926	100	82.4	0.0413	0.810 to 0.983

## Data Availability

The datasets generated and/or analyzed during the current study are not publicly available due to the protection of patient privacy but are available from the corresponding author at a reasonable request.

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
