# Peer review of "Predicting a Favorable (mRS 0–2) or Unfavorable (mRS 3–6) Stroke Outcome by Arterial Spin Labeling and Amide Proton Transfer Imaging in Post-Thrombolysis Stroke Patients"

_jpm, 2023, doi:10.3390/jpm13020248_

Round 1

Reviewer 1 Report

He et al. conducted a study in order to evaluate the ability of some MRI sequences (i.e. Arterial Spin Labeling and amide proton transfer) of predicting the stroke outcome at discharge, assessed through the mRS. The topic is interesting, but the manuscript requires to be reviewed before publication. In particular, statistical analysis should be improved.

Title: the authors should define mRS.

Abstract: 

1.     mRS is the acronym of modified Rankin Scale, not of “modified Rankin Scores”. Please correct.

2.     Did the authors consider the 3-month mRS or the discharge one? Please specify this point in the abstract.

3.     Which kind of neuroradiological examination has been considered in the study? How much time has been passed between the ischemic stroke onset and the neuroradiological examination considered in the study? Please specify.

Introduction

1.     The introduction needs to be checked for grammar error.

2.     The time window to perform thrombolysis after ischemic stroke onset is 4.5 hours, which may be extended to 9 hours if the neuroradiological exams detect the presence of savable tissue. The time interval of 6 hours is, on the other hand, considered for mechanical thrombectomy. Please correct.

3.     FLAIR needs to be defined only one time.

4.     Traditionally, these images have been used to extrapolate the eventual outcomes of ischemic stroke.”. Please clarify the meaning of this sentence. Did the authors mean that the dimension of the ischemic lesion calculated basing on the FLAIR sequences could predict the outcome or did the authors mean something else? Moreover, a reference is needed in this point.

Materials and methods

1.     Was the informed consent an inclusion criterion? The authors should include “undergoing thrombolytic therapy” as an inclusion criterion for the study participation.

2.     Please clarify whether some patients underwent mechanical thrombectomy or if these patients have been excluded from study participation.

3.     Did the authors perform a sample size calculation?

Results

1.     Please revise the sentence “Early to collect 73 patients…”

2.     What does metabolic arthritis means? Please specify.

3.     Please add some information regarding the stroke characteristics of the study population, such as the pathogenic mechanisms of stroke in the selected population according to the TOAST classification, the proportion of stroke involving the anterior and the posterior circulation, the mean+/- SD of the stroke dimension. Moreover, the authors should report the mean +/- SD of the mRS before the stroke onset and the results of the statistical comparison of this parameter between the two study groups.

4.     The authors should perform a multivariate statistical comparison including other factors which may influence the stroke outcome, such as the previous mRS, the NIHSS at onset (which resulted to be different in the two subgroups in the univariate analysis), the dimension of the stroke lesion, and so on.

5.     Please specify the mean time +/- SD passed between the stroke onset and the radiological examination considered in the study. 

Discussion

1.      “… in the stroked region of patients with mild brain damage.” The meaning of this sentence is not clear. Please revise.

In conclusion, the topic of the manuscript is interesting and the manuscript is easy to read, but substantial improvements are needed for publication.

Author Response

请参阅附件。

Reviewer 2 Report

The study is well conducted. The manuscript needs good grammatical correction (e.g. line no 164). 

Reviewer 3 Report

Dear authors,

Your manuscript is potentially benefit for stroke service. You were reporting about the use of some special MRI techniques to predict the favorable outcome (mRs 0-2) of thrombolytic stroke treatment, which, to my understanding, included the cases with post thrombolytic treatment. I would like to say that if it will be clearly defined, esp the inclusion criteria, I think it will be more useful for readers to apply the findings in their real stroke practice. Please consider my comments as below:

A. General comments

The authors aimed to quoted the benefit of the advanced MRI techniques in predicting post thrombolytic treatment outcomes. However, I think the different types of stroke, i.e. small vessel or large vessel stroke usually results in different outcome by its own nature. Anterior vs. posterior circulation stroke brings about different outcomes. Time of stroke onset (or time of last seen well) to treatment initiation also influences the outcome. In addition, stroke without complications (such as hemorrhagic transformation or massive brain edema) commonly has a bettter outcome. When the initial neurological presentations are tested comparable between the two different outcome groups, that I think the aims of this study will be met. 

I wonder whether the authors intended to assess the immediate stroke outcomes which was just 1 week after the treatment. Because stroke outcomes usually are assessed at 30 or 90 days after stroke onset.  This point needs clearly be defined because it was not clearly mentioned in the manuscript. If it was not evaluated as usual time, so, please address the reason or benefit of the current assess time.

Additionally, What the new knowledge or good practice beyond the stroke guideline used in your country or internationally can be extracted from the current research study? This should be emphasized in regards with the time- based approach of the current stroke therapy guideline.

B. Specific comments by manuscript sections

1. The title. It seems somewhat not cover or reflect the overall feature of the study. Since the patients in this study were acute stroke patients who received intravenous thrombolysis, and the aim of this study was to test the imagings whether they are effective in differentiation between the patients with favorable (usually means mRs 0-2) or unfavorable (mRs>3-6) outcomes after thrombolysis. I would like to suggest like this "Predicting a favorable stroke outcome by arterial spin labeling and amide proton transfer imaging in post-thrombolysis stroke patients"  

2. Introduction. The significance and usage of ASL and APT imaging in "selecting acute stroke patients" who will benefit from acute treatment, or "prognosticating of levels of recovery" in the previous studies should be much broadly reviewed (paragraph 2,3). 

Also, the gap of knowledge and usage of the MRI techniques used in this study should be stressed. What are the advantages or disadvantages of these MRI tecniques compared with DWI-PWI mismatch study, or cerebral blood flow (CBF) and cerebral blood volume (CBV) studies previously mentioned in acute stroke treatment.

The content in line 49-52 may lead to some misunderstandings.

-mRs can be regarded as an objective assessment tool, it is not subjective but a bit superficial evaluation because of its overall evaluation in nature.

-How does "socio-economic status" affect the mRs as stated? Please clarify. I guess the authors intended to say it indirectly affect the mRs.

-NIHHS is used for stroke severity assessment, while mRs is for stroke outcomes assessment usually done in 30 and 90 days after stroke onset in most of the stroke studies. The content in line 93-96 correctly explains the usage of the two scores.

3. Methods

The term "subacute phase ischemic stroke" is better to represent the subjects in this study. Please avoid the term " subacute ischemic stroke" which means the stroke patients who come to receive a specific treatment beyond the recommended golden period for the treatments. Please recheck this terms throughout the manuscript including key words of the abstarct.

Please clarify the inclusion criteria 

-Item (2), I guess the authors intended to state that the patients with no previous strokes or those with good recovery after the strokes (and then please define the mRs as indicated for the study, e.g. 0-1 or others).

-Item 4. What dose the "the onset time of stroke" mean? Does it mean the time from the stroke onset to any treatments, or to undergo the ASL and APT imaging in this study? Please clarify. Duration of stroke appearance has effect on the volume of brain tissue loss and its prenumbra.

In the exclusion criteria

-Item (4). Please specifiy clearly the "concurrent neurological disorders" that were excluded from the study, wheter this refered to Parkinson's disease, epilepsy, migraine, etc.

-The authors did not mention about how to calculate the sample size needed in this study.

3. Results.

-Could the authors please detail more about the study patients' characteristics, which may influence the outcome, such as cardiovascular or other systemic disease as the comorbidities. Also, the authors should explain what " metabolic arthritis" means?

-How many patients and by what reasons that they were excluded from the study patients initially enrolled should be declared.

-Table 1 provides no more information from the description text before it, It can be deleted

-Table 3 and 4 should be merged into one table which the p values were inserted following paired columns of ASL and APT of good vs. poor stroke outcomes.

4. Discussion

The authors described the basic and physical principles of the techniques well, however, how do the findings correspond to the neurological outcomes reported in the previous studies similar to this one. To draw a conclusion of the correlation between neuro-radiological findings and neurological outcomes may be more useful to the readers and will highlight this study results. The authors plesae discuss and add more details.

Can the selection of ROIs by the two neuroradiologists, who were clinically -blinded both pre and post thrombolyis periods, effect the results, or the authors had designed the preventive measures for this already? The authors please clarify.

Is it possible to apply the MRI tecniques in this study for pre-thrombolytic selection of the stroke patients who will gain clinical benefit from thrombolytic therapy. My suggestion is the authors should discuss or comment on this issue. 

5. Conclusion

For the last phrase of this section and also in the abstract ( and therapy........... high-risk patients), I guess the authors didn't mean the prethrombolysis selection of suitable patients for the treatment, and what is the term " high-risk patients" refer to, i.e. high-risk for severe disability, fixed handicap, or etc.

Please see the authors' original submitted manuscript attacched with highlight marks.

Round 2

Reviewer 1 Report

The authors improved the manuscript.

Unfortunately, I continue to have some concerns:

-          In the title outcome should be singular, not plural.

-          In the inclusion criteria, what does “a good recovery after the strokes” mean? A mRS<3?

-          The authors should include mechanical thrombectomy within the exclusion criteria.

-          Results: “pre-circulation and post-criculation strokes” makes no sense. The authors should instead write “anterior circulation and posterior circulation strokes”.

-          “According to the ischemic stroke TOAST etiology classification, we analyzed the 220 patients in this study and found that most patients had aortic atherosclerosis”: did the authors mean carotid atherosclerosis? Actually, there are no information regarding the proportion of stroke subtypes in the study population. This information has a great relevance since, for example, lacunar strokes present very much differences from cardioembolic and atherosclerotic strokes, including different outcome. The authors should specify this point or, alternatively, report this lack of information as a limitation of the study.

-          The part of the Material and Methods concerning the statical analysis should be moved from the heading “Image Analysis” to a separate heading (“Statistical analysis”). Furthermore, no clear explanation is available on the statistical test employed for sample size calculation. Finally, the sample size calculation should also be moved to the methods, in the section “Statistical analysis” (it is actually in the results).

-          The multivariate logistic regression, according to what reported in the results, has been performed only including the “imaging markers” which differed significantly between groups in the univariate comparison, but all clinical variables which differed significantly between the poor prognosis and the good prognosis groups in the Mann-Withney U-test (NIHSS at admission, mRS at admission) have not been considered. Please, run another logistic regression including these variables, or give a good explanation for not considering them.

-          “The ASL and APT parameters are significantly different from the NIHSS on admission, the mRS on admission and the dimension of the stroke lesion.”: The meaning of this sentence is not clear. Please specify.

-          Table 4: No “*” are visible in the table, near to the significant p value. Please add it.

-          The English language has been improved, but some sections (especially the Matherials and methods one) should still be revised.

Author Response

Dear Editors and Reviewers:

      Thank you for your letter and for the reviewers’ comments concerning our manuscript entitled Differentiating Patients with Low and High mRS Scores of Ischemic Stroke Using Arterial Spin Labeling and Amide Proton Transfer Imaging. Those comments are all valuable and very helpful for revising and improving our paper. We have studied the comments carefully and have made corrections which we hope meet with approval. The main corrections in the paper and the responses to the reviewer's comments are as flowing:

Reviewer 1

Question1: In the title outcome should be singular, not plural.

Response1: Thanks for your careful checks. In our resubmitted manuscript, we have revised the title.

Original: “Predicting a favorable (mRS 0-2) or unfavorable (mRS 3-6) stroke outcomes by arterial spin labeling and amide proton transfer imaging in post-thrombolysis stroke patients”

Revised: “Predicting a favorable stroke outcome by arterial spin labeling and amide proton transfer imaging in post-thrombolysis stroke patients” 

Question 2: In the inclusion criteria, what does “a good recovery after the strokes” mean? A mRS<3?

Response 2: Thanks for your careful reading. We want to say people who have recovered almost completely after a previous stroke.

Original: “a good recovery after the strokes”

Revised: “those with almost completely recovery after the a previous stroke” 

Question 3: The authors should include mechanical thrombectomy within the exclusion criteria.

Response 3: Thanks for your comments. We have added the exclusion criteria (5).

Undergoing mechanical thrombectomy therapy

Question 4: Results: “pre-circulation and post-criculation strokes” makes no sense. The authors should instead write “anterior circulation and posterior circulation strokes”

Response 4: Thanks for your careful reading. Thanks for your careful reading. As suggested by the reviewer, we have corrected “pre-circulation and post-criculation strokes” into “anterior circulation and posterior circulation strokes”.

Question5: “According to the ischemic stroke TOAST etiology classification, we analyzed the 220 patients in this study and found that most patients had aortic atherosclerosis”: did the authors mean carotid atherosclerosis? Actually, there are no information regarding the proportion of stroke subtypes in the study population. This information has a great relevance since, for example, lacunar strokes present very much differences from cardioembolic and atherosclerotic strokes, including different outcome. The authors should specify this point or, alternatively, report this lack of information as a limitation of the study.

Response 5: -Thanks for your comments. No classification studies were conducted to classify patients according to the etiology of TOAST in ischaemic stroke. We hope to add this to future studies. We have illustrated this in the limitations section of the article.

Question 6: The part of the Material and Methods concerning the statical analysis should be moved from the heading “Image Analysis” to a separate heading (“Statistical analysis”). Furthermore, no clear explanation is available on the statistical test employed for sample size calculation. Finally, the sample size calculation should also be moved to the methods, in the section “Statistical analysis” (it is actually in the results).

Response 6: -Thanks for your suggestion.

①We have moved the section on materials and methods dealing with static analysis from the heading "Image Analysis" to a separate title ("Statistical Analysis").

② Thanks for your careful reading. We have explained the statistical tests used for the sample size calculations.

③We have moved the sample size calculation to the methods in the "Statistical Analysis" section.

Question7: The multivariate logistic regression, according to what reported in the results, has been performed only including the “imaging markers” which differed significantly between groups in the univariate comparison, but all clinical variables which differed significantly between the poor prognosis and the good prognosis groups in the Mann-Withney U-test (NIHSS at admission, mRS at admission) have not been considered. Please, run another logistic regression including these variables, or give a good explanation for not considering them.

Response 7: - We were sorry for our carelessness. We have revised the table 4. We have made multivariate statistical comparisons that may affect stroke outcomes,including NIHSS admissionmRS (admission)  the dimension of the stroke lesion.

Question8: “The ASL and APT parameters are significantly different from the NIHSS on admission, the mRS on admission and the dimension of the stroke lesion.”: The meaning of this sentence is not clear. Please specify.

Response8: -Thanks for your comments.

Original: “The ASL and APT parameters are significantly different from the NIHSS on admission, the mRS on admission and the dimension of the stroke lesion.”

Revised: “There was a statistically significant relationship between NIHSSadmission, mRS (admission), and the dimension of the stroke lesion.”

Question9: Table 4: No “*” are visible in the table, near to the significant p value. Please add it.

Response9: -We feel so sorry for our careless mistakes. Thank you for your reminder. We have revised the table 4.

Question10: The English language has been improved, but some sections (especially the Matherials and methods one) should still be revised.

Response10: -Thanks for your comments. We have tried our best to polish the language in the revised manuscript.

We tried our best to improve the manuscript and made some changes in the manuscript.

We appreciate Editors'/Reviewers’warmwork earnestly and hope that the correction will meet with approval. Once again thank you very much for your comments and suggestions.

Reviewer 3 Report

Dear authors,

Thank you for your revision. I perceived your full effort in revision of the original manuscript. I would like to add some further minor comments for improving the readability and academic useful of the manuscript as follow:

1. I think the second followed by the initial part of the third paragraph of the discussion talked mostly the pathophysiology of microenvironment changes  (i.e. the acidity of brain tissue) of ischemic brain tissue, but lack of refering to some studies that correlated the different imaging techniques or findings, either single or combined techniques or findings, with the stroke outcomes exactly, which I think the information may support and enhance the results of the current study. For the basic radiological physics or biochemical theories, the authors can shortly brief them to a more concise content to avoid the diluation of significant main content to be addressed.

My initial comment in the first version of the submitted manuscript intended to have the authors to refer to the existing published studies reporting the  correlation between imaging findings and stroke outcomes.

2. If "Metabolic arthritis " was meant abnormal purine metabolism that led to arthritis as described, It should be "gouty arthropathy" , please consider.

3.Line 404-405. "Clinically, head MR is usually performed after thrombolysis to understand post-thrombolytic recanalization and to guide ischemic stroke reperfusion therapy."

   It is not clear whether the authors intended to mean that head MRI was clinically used to guide further reperfusion treatment (such as mechanical thrombectomy, ultrsound thrombolysis, etc.) after failed thrombolysis, or else. If so, I think a reference is required.

4. The title. I would like to suggest : "Predicting a favorable stroke outcome by...    ", and no need to add (mRs 0-2), since this is a standard classification of stroke outcomes used in most stroke studies.

5. Please carefully recheck that some abbreviations already have their full- written terms before the use of them in the following content.

Thank you.
